# High Substitution Synthesis of Carboxymethyl Chitosan for Properties Improvement of Carboxymethyl Chitosan Films Depending on Particle Sizes

**DOI:** 10.3390/molecules26196013

**Published:** 2021-10-03

**Authors:** Sarinthip Thanakkasaranee, Kittisak Jantanasakulwong, Yuthana Phimolsiripol, Noppol Leksawasdi, Phisit Seesuriyachan, Thanongsak Chaiyaso, Pensak Jantrawut, Warintorn Ruksiriwanich, Sarana Rose Sommano, Winita Punyodom, Alissara Reungsang, Thi Minh Phuong Ngo, Parichat Thipchai, Wirongrong Tongdeesoontorn, Pornchai Rachtanapun

**Affiliations:** 1Faculty of Agro-Industry, School of Agro-Industry, Chiang Mai University, Chiang Mai 50100, Thailand; s.thanakkasaranee@gmail.com (S.T.); jantanasakulwong.k@gmail.com (K.J.); yuthana.p@cmu.ac.th (Y.P.); noppol@hotmail.com (N.L.); phisit.s@cmu.ac.th (P.S.); thachaiyaso@hotmail.com (T.C.); 2The Cluster of Agro Bio-Circular-Green Industry (Agro BCG), Chiang Mai University, Chiang Mai 50100, Thailand; 3Center of Excellence in Materials Science and Technology, Chiang Mai University, Chiang Mai 50200, Thailand; pensak.amuamu@gmail.com (P.J.); warintorn.ruksiri@cmu.ac.th (W.R.); sarana.s@cmu.ac.th (S.R.S.); winitacmu@gmail.com (W.P.); 4Department of Pharmaceutical Sciences, Faculty of Pharmacy, Chiang Mai University, Chiang Mai 50200, Thailand; 5Plant Bioactive Compound Laboratory (BAC), Department of Plant and Soil Sciences, Faculty of Agriculture, Chiang Mai University, Chiang Mai 50200, Thailand; 6Department of Chemistry, Faculty of Science, Chiang Mai University, Chiang Mai 50200, Thailand; 7Department of Biotechnology, Faculty of Technology, Khon Kaen University, Khon Kaen 40002, Thailand; alissara@kku.ac.th; 8Research Group for Development of Microbial Hydrogen Production Process, Khon Kaen University, Khon Kaen 40002, Thailand; 9Academy of Science, Royal Society of Thailand, Bangkok 10300, Thailand; 10Department of Chemical Technology and Environment, The University of Danang—University of Technology and Education, Danang 550000, Vietnam; ntmphuong@ute.udn.vn; 11Doctor of Philosophy Program in Nanoscience and Nanotechnology (International Program/Interdisciplinary), Faculty of Science, Chiang Mai University, Chiang Mai 50200, Thailand; parichat_thi@cmu.ac.th; 12School of Agro-Industry, Mae Fah Luang University, Chiang Rai 57100, Thailand; wirongrong.ton@mfu.ac.th; 13Research Group of Innovative Food Packaging and Biomaterials Unit, Mae Fah Luang University, Chiang Rai 57100, Thailand

**Keywords:** chitosan, carboxymethyl chitosan, particle size, water solubility, degree of substitution

## Abstract

This study investigated the effect of chitosan particle sizes on the properties of carboxymethyl chitosan (CMCh) powders and films. Chitosan powders with different particle sizes (75, 125, 250, 450 and 850 µm) were used to synthesize the CMCh powders. The yield, degree of substitution (DS), and water solubility of the CMCh powders were then determined. The CMCh films prepared with CMCh based on chitosan with different particle sizes were fabricated by a solution casting technique. The water solubility, mechanical properties, and water vapor transmission rate (WVTR) of the CMCh films were measured. As the chitosan particle size decreased, the yield, DS, and water solubility of the synthesized CMCh powders increased. The increase in water solubility was due to an increase in the polarity of the CMCh powder, from a higher conversion of chitosan into CMCh. In addition, the higher conversion of chitosan was also related to a higher surface area in the substitution reaction provided by chitosan powder with a smaller particle size. As the particle size of chitosan decreased, the tensile strength, elongation at break, and WVTR of the CMCh films increased. This study demonstrated that a greater improvement in water solubility of the CMCh powders and films can be achieved by using chitosan powder with a smaller size.

## 1. Introduction

Currently, demands for the development of new functional materials with biocompatibility and biodegradability are continuously increasing, which has led to an increase in the utilization of biopolymers as compared to synthetic conventional polymers [1,2].

Among biopolymers, chitosan is an attractive material, owing to its biocompatibility, antimicrobial activity, cheapness, and low toxicity. It has been widely used in the pharmaceutical industry (e.g., drug delivery carrier) [3,4], for biomedical applications (e.g., wound dressing) [5,6], the cosmetic industry (e.g., encapsulation of essential oils) [7], and the food packaging industry (e.g., antimicrobial film) [8]. Chitosan is a biopolymer in the category of polysaccharides, and comprises glucosamine and N-acetylglucosamine linked by β (1–4) bonds. Chitosan is obtained by the deacetylation of chitin, for which the abundant renewable sources are the shells of crustaceans and mollusks [3,9]. However, chitosan is insoluble in neutral water (pH ~7) owing to the presence of amino groups (–NH_2_) in the chitosan, in which this functional group remains unprotonated in neutral water [10]. Therefore, an improvement of water solubility of chitosan is required prior to its application.

The conversion of chitosan into carboxymethyl chitosan (CMCh) is a facile and alternative means to address the unsatisfied insoluble chitosan, in which the chemical structure of chitosan becomes more polar. Therefore, the CMCh can be dissolved in water over a wide range of pH, which provides a convenient use of CMCh in various applications [11]. In addition, CMCh has antimicrobial activity, biocompatibility, and low toxicity, which has been incorporated into biopolymers (e.g., starch and cellulose) to enhance the mechanical/thermal properties and antimicrobial activity of blend films in the development of active food packaging [12,13,14,15]. Moreover, CMCh has attracted considerable attention as a tool for postharvest life extension by CMCh coating [16]; a fixative for Eau de Cologne [17]; and for blend films with soy protein isolate [18], rice starch [19], antioxidant and moistening materials [20], and edible inks [21].

The CMCh can be synthesized in sodium hydroxide (NaOH) solution, and the effect of NaOH solution with different concentrations on the degree of substitution (DS) has been studied by Abou-Zied et al. [22]. They reported that the DS increased with respect to the concentration of the NaOH solution up to 50% (*w/v*), but it decreased at above 60% (*w/v*) of NaOH solution [22]. Furthermore, the properties of synthesized carboxymethyl cellulose prepared from the cellulose with different particle sizes were investigated by Yeasmin and Modal [23]. They found that DS, yield, viscosity, and water solubility increased as the cellulose particle size decreased [23]. In addition, Chaiwong et al. reported that water solubility of CMCh was dependent on the molecular weight of chitosan used as a starting material. That is, the water solubility of CMCh increased with the decreasing molecular weight of chitosan [20].

In this context, we attempted to study the effect of chitosan particle sizes on the yield, DS, water solubility, and change in a reactive functional group of the synthesized CMCh, which have not yet been investigated elsewhere. In addition, we also determined the water solubility, mechanical properties, and water vapor transmission rate (WVTR) of the CMCh films.

## 2. Results and Discussion

### 2.1. Chemical Structure

The FT-IR spectra of chitosan and the CMCh powders obtained from various chitosan particle sizes are shown in Figure 1. The characteristic peaks of chitosan powder were observed at 3000–3600, 2879, 1599, 1401, and 1160–1000 cm^−1^, which were attributed to –OH stretching, C–H stretching, N–H bending, C–H bending of CH_2_ group, and C–O stretching, respectively [24,25]. The absorption bands of CMCh powders obtained from chitosan with higher particle sizes (75–850 µm) were relatively similar to that of chitosan. The intensity of these characteristic bands clearly increased with the decrease in chitosan particle size, indicating an increase in reactive functional groups (e.g., –OH and –NH_2_ groups) in the CMCh powders. Particularly, the observation of strong peak intensity in the vicinity of 1599 cm^−1^ might correspond to the vibration of the COO– group overlapped with the original N–H bond of chitosan, which might imply that there was some conversion of chitosan to carboxymethyl chitosan. Notably, a new peak at 1741 cm^−1^ was detected in the spectrum of CMCh 75 µm, which is related to the C=O stretch of carboxylic group [12,26,27]. This confirms that the chitosan with a smaller particle size was highly converted to carboxymethyl chitosan.

### 2.2. Morphology of Chitosan and CMCh Powders

The morphology of chitosan and CMCh chitosan powders with different particle sizes was analyzed using the stereo microscope and SEM. As shown in Figure 2, chitosan powders are relatively opaque, and have a slightly rough surface and irregular shape particles. Conversely, CMCh powders are relatively transparent due to the reduction in crystallinity after chemical modification, which is in agreement with our previous study by XRD result [12]. In addition, CMCh powders have rougher surfaces compared to chitosan. Likewise, SEM images (Figure 3a,b) showed that the surface morphology of CMCh powders was rougher than that of chitosan powers. This was due to the damage on chitosan surfaces and the weakening of chitosan structures, which caused the reduction in chitosan crystallinity [28,29]. This allowed the carboxymethylization, and resulted in the formation of the bulky groups (–CH_2_COOH) on chitosan surfaces, which is consistent with the studies of Mohamed et al. [30] and Sabaa et al. [31]. The average particle sizes of chitosan and CMCh powders were measured by SEM and are summarized in Appendix A. The width of CMCh particles was greater than that of chitosan particles, due to the effects of the formation of the bulky groups (–CH_2_COOH) on their surfaces. In contrast, the length of CMCh particles was smaller than that of chitosan particles due to the effects of the chemical reaction during surface modification, which led to chain scission.

### 2.3. Yield of CMCh Powders

The yield of CMCh powders prepared from chitosan powder with different particle sizes is shown in Figure 4. Notably, the conversion of chitosan into CMCh was strongly dependent on chitosan particle sizes. As the particle size decreased, the yield of CMCh powders increased, indicating an increase in the content of CMCh. This was related to the surface areas of the chitosan used in the synthesis, in which the chitosan with smaller particle sizes provided greater surface areas. An increase in the surface area of chitosan caused potential collisions between chitosan particles and reactants, which resulted in an increase in the yield of CMCh synthesized from the chitosan with smaller particle sizes. This phenomenon is consistent with the investigations of Rahman et al. [32] and Yeasmin and Mondal [23], who found that the yield of carboxymethyl cellulose (CMC) increased with the decreases in cellulose particle size. In addition, some research groups reported that the yield of CMCh was also dependent on the solvent, concentration of NaOH, reaction temperature [33], and molecular weight of chitosan [20].

### 2.4. DS and Water Solubility of CMCh Powders

The carboxymethylation of carbohydrate polymers (e.g., cellulose, starch, and chitosan) has been extensively investigated. This is due to its facile modification achieved through the acid-catalyzed reaction [34], in which the DS of carboxymethylation depends on the concentration of base [35,36,37], amount of acid [28], type of solvent [38], nature of polymeric materials [23], reaction temperature [39,40], reaction time [39,40], and microwave radiation (power) [41]. Due to a higher polarity, increases in DS generally result in improved water solubility in the biopolymers [42]. In this study, the effect of chitosan particle size on the DS and water solubility of CMCh is shown in Figure 5. Notably, the DS and water solubility of CMCh were dependent on the chitosan particle size. The CMCh powder with smaller particle sizes had a higher DS due to the larger surface areas generated by smaller particles, which led to a greater conversion of chitosan into CMCh. A similar influence of particle size on the DS of CMC has also been reported by Rahman et al. [32] and Yeasmin and Mondal [23]. The higher DS implied an increase in the polarity of the CMCh powders, as reflected by the larger number of polar groups (–OH, –NH_2_, and –COOH) and explained by the FT-IR results. Therefore, the CMCh prepared from the chitosan powder with smaller particle sizes easily dissolved in water, which led to a higher water solubility (from 88.9 to 95.5%). As a consequence, the DS and water solubility of the CMCh powder with smaller particle sizes were greater than the powder with larger particle sizes.

### 2.5. Morphology of CMCh Films

The fractured surface of the CMCh films that were prepared with CMCh based on chitosan powders with different particle sizes is shown in Figure 6. All CMCh films exhibited a continuous phase without internal pores. Notably, the roughness of the CMCh films increased as the chitosan particle size increased. This roughness could be attributed to the imperfect solubility of CMCh powders during the film forming process, which may result in the deterioration of mechanical properties when affected by a larger particle size and a lower surface area.

### 2.6. Mechanical Properties of CMCh Films

The mechanical properties of polymeric materials are dependent on several factors, including intermolecular forces (i.e., H-bond), molecular weight (chain length), and the degree of crystallinity (ordered packing arrangement) [43,44]. Tantala et al. [45] reported that the CMCh film prepared from the chitosan with higher molecular weight (polymer) has a higher tensile strength than those prepared from the chitosan with lower molecular weight (oligomer) [45]. In this study, the effect of chitosan particle size on the mechanical properties of the CMCh films is shown in Figure 7. Obviously, the tensile strength and elongation at break of the CMCh films were strongly dependent on the chitosan particle size. As expected, the CMCh film prepared from the chitosan with a smaller particle size showed a higher tensile strength and elongation at break. As mentioned earlier, the powder with smaller particle sizes had a higher surface area in the substitution reaction, which led to a higher conversion of chitosan to CMCh, as reflected by a higher DS. This resulted in an increase in the intermolecular force between molecules of CMCh [46], leading to an enhancement of their mechanical properties. Conversely, the crystallinity of the CMCh film decreased due to the chemical reaction (i.e., NaOH) and the formation of bulky groups (–CH_2_COOH) on the chitosan chains. Therefore, the CMCh prepared from the chitosan with a smaller particle size has a lower crystallinity than that of the CMCh film, owing to a higher number of bulky groups. This led to the enhancement of the conformation flexibility (i.e., elongation at break) of the CMCh films [47]. This result is consistent with Suriyatem et al. [12] who reported that the introduction of CMCh into rice starch (50%) increased the tensile strength and elongation at break of rice starch. In addition, this result is in agreement with Rachtanapun et al. [28] who found that the tensile strength and elongation at break of carboxymethyl bacterial cellulose film increased with DS and intermolecular force.

### 2.7. Contact Angle of CMCh Films

As shown in Figure 8, the dynamic water contact angle of the CMCh film was remarkably dependent on the chitosan particle size. As the chitosan particle size decreased, the water contact angle of CMCh films decreased. In addition, the water contact angle of all CMCh films decreased with time (from 0 to 20 s). This implies that the CMCh film prepared from CMCh-based chitosan powder with a smaller particle size has a more hydrophilic character than that of the CMCh film, due to the high number of COOH groups on the larger surface area of CMCh prepared from chitosan powder with a smaller particle size. This contact angle result is consistent with the FT-IR, DS, and water solubility of CMCh powders.

### 2.8. Water Solubility and Water Vapor Transmission Rate of CMCh Films

As shown in Figure 9, the chitosan particle size used to synthesis CMCh strongly affected the water solubility of the CMCh films. The water solubility of the CMCh films increased (94.3–97.9%) as the particle size of the chitosan used in the synthesis of the CMCh powder decreased. This was related to the polarity or the hydrophilicity of the CMCh films. Essentially, the polymeric films with a higher polarity have a higher chemical affinity with water molecules, such that water can be initially adsorbed on the surface and diffused into the polymeric films. Finally, the polymeric moieties of the films are dissolved. Therefore, when the CMCh film prepared from the chitosan with the smaller particle size (75 µm) was exposed to water, a larger amount of –COOH groups (on the surface of the CMCh film) and H-bonding (between the CMCh chains) became separated due to competition among the water molecules [48,49]. Accordingly, the CMCh film with a higher polarity (prepared from chitosan with a smaller particle size) was easily deformed and dissolved in water. This result is consistent with the FT-IR results, DS, and water solubility of the CMCh powders. In addition, this finding is also similar to that of the study of Tantala et al. [50], who stated that the water solubility of CMCh films was dependent on the DS, irrespective of the plasticizers. Notably, the water solubility of CMCh films was higher than that of the CMCh powders due to the influence of the hydrophilic character of the plasticizer (glycerol) containing the CMCh film.

Generally, water vapor permeation of polymeric films strongly depends on several factors, such as the polarity or hydrophilicity, intermolecular forces, and degree of crystallinity [43,51]. In addition, the molecular weight of chitosan affects the WVTR of the CMCh film, as the CMCh film prepared from the chitosan with higher molecular weight exhibits a lower WVTR compared to those prepared from the chitosan with a lower molecular weight [45]. In this study, the effect of chitosan particle sizes on the WVTR of CMCh films is illustrated in Figure 9. The CMCh film prepared from chitosan with the largest particle size (850 µm) exhibited the lowest WVTR value of 8.8 g/day·cm^3^. As the chitosan particle size decreased, the WVTR of the CMCh films increased from 8.8–18.3 g/day·cm^3^. The increase in the WVTR of the CMCh films is due to an increase in the hydrophilicity of the CMCh film, which is caused by the presence of abundant −COOH and −OH groups, as explained by the FT-IR, DS, and contact angle results. The water vapor molecule can easily react with such polar groups (dissociation) and penetrate into the CMCh films, because no CMCh films have any internal pores, as indicated in the SEM images (Figure 6). Therefore, the CMCh film prepared from CMCh with a higher DS exhibited a higher WVTR, which confirmed that the CMCh films had become more hydrophilic in character. This finding is consistent with the analysis of Rachtanapun et al. [52], who reported that the CMC films with a higher DS showed a higher WVTR.

## 3. Materials and Methods

### 3.1. Materials

Shrimp chitosan (molecular weight of 1000–1500 kDa) was purchased from Ta Ming Enterprises Co., Ltd. (Samut Sakhon, Thailand). Acetic acid glacial anhydrous and sodium hydroxide (NaOH) were purchased from Merck & Co., Inc. (Darmstadt, Germany). Ethanol and methanol were purchased from Northern chemical Co., Ltd. (Chiang Mai, Thailand). Magnesium nitrate (Mg(NO_3_)_2_) was purchased from Renkem chemical (Bangalore, India). Monochloroacetic acid was purchased from Sigma-Aldrich (Burlington, MA, USA).

### 3.2. Synthesis of CMCh Powders

Prior to the synthesis of CMCh powder, chitosan was ground and sieved using a Retsch ZM 200 Ultra Centrifugal Mill (Endecotts, London, UK) to obtain various particle sizes of 75, 125, 250, 425, and 850 µm. Approximately 25 g of the obtained chitosan was individually suspended in 400 mL of 50% (*w/v*) sodium hydroxide solution. Then, 100 mL of isopropanol was added, and the chitosan solution was stirred at 50 °C for 1 h. Next, the solution of monochloroacetic acid (in isopropanol) was gradually added into the chitosan solution. This mixture was transferred into an ED 56 drying and heating chamber (Binder Co., Tuttlingen, Germany) and heated at 50 °C for 4 h. The solid part of the mixture was separated by filtering, and 250 mL of 70% (*v*/*v*) methanol was added into the solid part and stirred for 10 min. Meanwhile, the pH of this mixture was adjusted to 7 by adding acetic acid prior to filtering. This washing process was performed 5 times. Lastly, 250 mL of 95% (*v*/*v*) methanol was added into the solid part and stirred for 10 min. The mixed solution was finally filtered, and the solid part was dried at 80 °C for 12 h in a drying and heating chamber. The yield of the obtained CMCh powder was calculated using Equation (1):(1)$$Yield\;\left(\%\right)=\frac{W_{1}}{W_{0}}\times 100\;$$
where *W*_0_ is the weight of chitosan (g), and *W*_1_ is the weight of the obtained CMCh (g).

### 3.3. Preparation of CMCh Films

To prepare the CMCh films, 3 g of the obtained CMCh from chitosan with different particle sizes was individually added to 100 mL of DI water and stirred at 80 °C for 10 min. Then, an optimal content of 30% (*w*/*v*) of glycerol was added in order to improve the brittleness of the CMCh films. The obtained CMCh solution was cast on the plate and dried at 40 °C for 24 h. The as-prepared CMCh films were stored in a chamber of 52 ± 1% RH [Mg(NO_3_)26H_2_O] at 25 ± 1 °C prior to analysis.

### 3.4. Characterizations

#### 3.4.1. Infrared Spectroscopy

To determine the functional groups of chitosan and CMCh powders, a Frontier Fourier transform infrared (FT-IR) spectrophotometer (PerkinElmer, Waltham, MA, USA) was used. The spectra of samples were recorded from 3950 to 450 cm^−1^.

#### 3.4.2. Morphology of Powders

To study the morphology of chitosan and CMCh powders, a JSM 5910 L LV-Scanning Electron Microscope (SEM) (JEOL Ltd., Boston, MA, USA), which has an accelerating voltage of 10 kV, and a Leica S8 APO Greenough stereo microscope (Leica Microsystems, Wetzlar, Germany) were used.

#### 3.4.3. DS

The functional DS of CMCh was determined using a titration technique. Firstly, 0.2 g of the obtained CMCh was added to 40 mL of DI water and stirred to obtain a homogeneous solution. Then, the pH of the CMCh solution was adjusted to ≤2 by adding hydrochloric acid. Next, the CMCh solution was titrated with 0.1 M sodium hydroxide until reaching the equivalent point. Finally, the DS of CMCh was calculated using Equation (2):(2)$$DS=\frac{161\times A}{M_{CMCh}-\left(58\times A\right)}$$
where *M_CMCh_* is the mass of *CMCh* (g); A was calculated using Equation (3):(3)$$A=V_{NaOH}\times C_{NaOH}$$
where *V_NaOH_* is the volume of NaOH (mL) used in titration; *C_NaOH_* is the concentration of NaOH (M).

#### 3.4.4. Water Solubility of CMCh Powders

To measure the water solubility of the obtained CMCh powders, 0.1 g of CMCh was added into 10 mL of DI water and stirred at 25 °C for 30 min. Then, the CMCh mixture was filtered and dried at 105 °C for 24 h. The dried residue was weighed, and the water solubility of CMCh powder was calculated using Equation (4):(4)$$Water\;solubility\;\left(\%\right)=\frac{W_{0}-W_{1}}{W_{0}}\times 100$$
where *W*_0_ is the initial weight of CMCh powder; *W*_1_ is the final weight of CMCh powder (g).

#### 3.4.5. Morphology of CMCh Films

To study the morphology and the internal structure of CMCh films, a JSM-IT300 Scanning Electron Microscope (JEOL Ltd., Tokyo, Japan) was used. Prior to SEM analysis, the film was fractured in liquid nitrogen and vacuum-sputtered with gold.

#### 3.4.6. Mechanical Properties of CMCh Films

A H1KS Universal Testing Machine (Tinius Olsen, Horsham, PA, USA) was used to investigate the tensile strength and elongation at break of the CMCh films. Ten specimens of each film were prepared in the form of a rectangular strip (1.5 × 15 cm) and stored at 50% RH and 23 ± 2 °C for 24 h prior to measurement, according to ASTM D882-10.

#### 3.4.7. Water Solubility of CMCh Films

To investigate the water solubility of CMCh films, three specimens of each film were prepared in the form of a square shape (1 × 1 cm). The specimen was weighed, and stored at 50% RH and 23 ± 2 °C for 7 d. After that, the specimen was added to 50 mL of DI water and agitated for 2 h using a shaker. The residue of CMCh film was separated and dried at 105 °C for 24 h, according to ASTM D618-05. Then, the dried residue was weighed, and the water solubility of CMCh films was calculated using Equation (5):(5)$$Water\;solubility\;\left(\%\right)=\;\frac{W_{0}-W_{1}}{W_{0}}\times 100$$
where *W*_0_ is the initial weight of the CMCh film; *W*_1_ is the final weight of the CMCh film (g).

#### 3.4.8. Contact Angle of CMCh Films

To analyze the hydrophilic properties of CMCh films, the dynamic contact angle of CMCh films was measured using a DSA30B Drop Shape Analyzer (KRÜSS, Hamburg, Germany) with sessile water drops (volume 10 μL). The water contact angles of the films were recorded at 0, 5, 10, 15, and 20 s, and the water contact angle values of each sample were calculated as the average of five measurements.

#### 3.4.9. WVTR

To investigate the WVTR of the CMCh films, the WVTR test was performed according to ASTM E96-93. Three specimens of each film were prepared in circular shapes with a diameter of 8 cm and placed on circular aluminum cups that each contained 10 g of dried silica gel. Then, each specimen was sealed with the cup using paraffin wax, and the as-prepared cups were weighed and stored at 50% RH and 25 °C for 7 d, during which the cups were weighed every day to obtain a slope of weight gain (*y*-axis) against time (*y*-axis). Finally, the *WVTR* was calculated using Equation (6):
(6)$$WVTR=\frac{Slope}{Film\;area}$$


## 4. Conclusions

The present study showed that the chitosan particle size strongly affected the chemical and physical properties of the synthesized CMCh powders and the CMCh films. As the chitosan particle size decreased, the DS, yield, and water solubility of the CMCh powders increased. These were related to the increase in polarity of the CMCh powder caused by the larger surface area of the chitosan with a smaller particle size, which was used in the synthesis of CMCh powder. The CMCh powder, synthesized from chitosan with a smaller particle size, has a larger carboxyl group (confirmed by FT-IR result), which resulted in the enhancement of hydrophilicity. Likewise, the water solubility and WVTR of CMCh films increased as the chitosan particle size decreased. Moreover, the tensile strength and elongation at break also increased, owing to an increase in intermolecular forces between the CMCh chains. This study provided a better understanding of the synthesis of water-soluble CMCh powder and the facile preparation of CMCh films of varying particle sizes of chitosan, as well as a meaningful insight into the potential uses of CMCh in the development of novel functional materials for the biomedical, pharmaceutical, and food packaging industries.

## Figures and Tables

**Figure 1 molecules-26-06013-f001:**
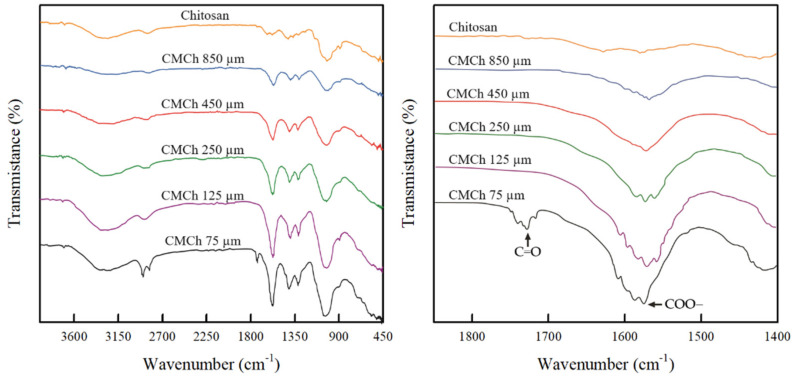
The FT−IR spectra of chitosan and CMCh powders prepared from chitosan with different particle sizes.

**Figure 2 molecules-26-06013-f002:**
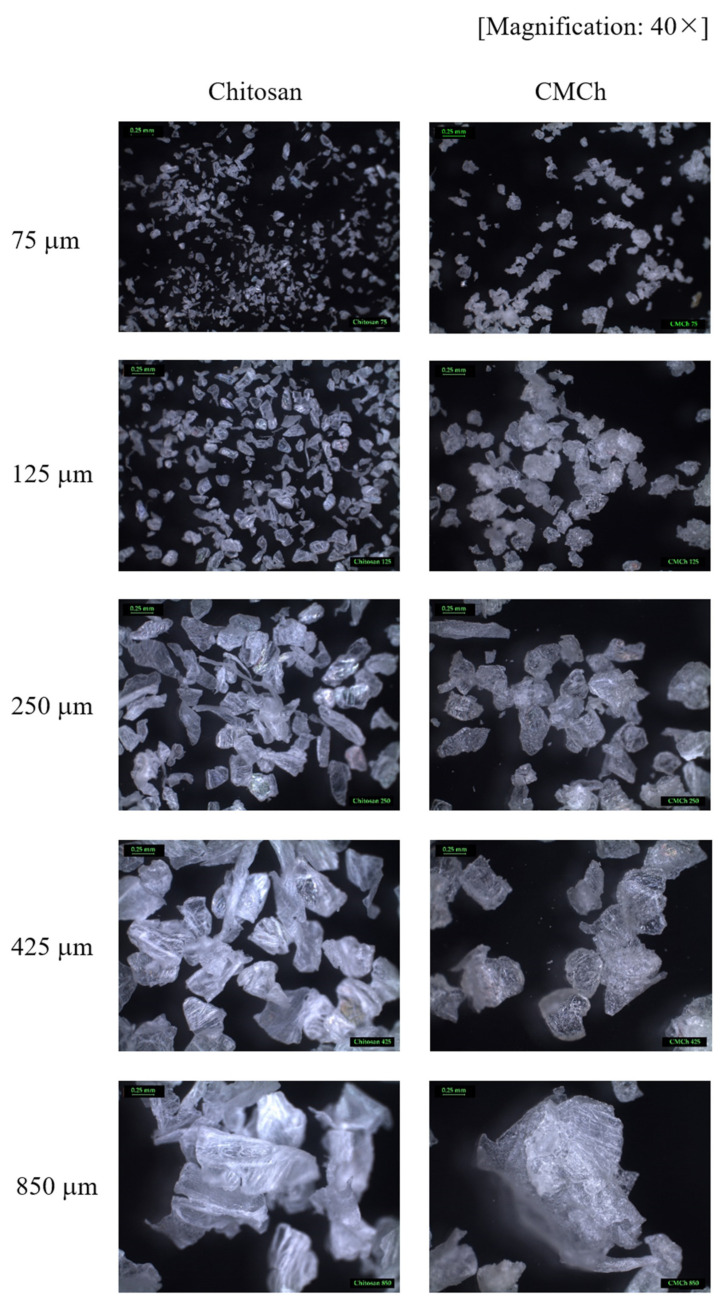
The morphology and physical appearance of chitosan and CMCh powders prepared from chitosan with different particle sizes.

**Figure 3 molecules-26-06013-f003:**
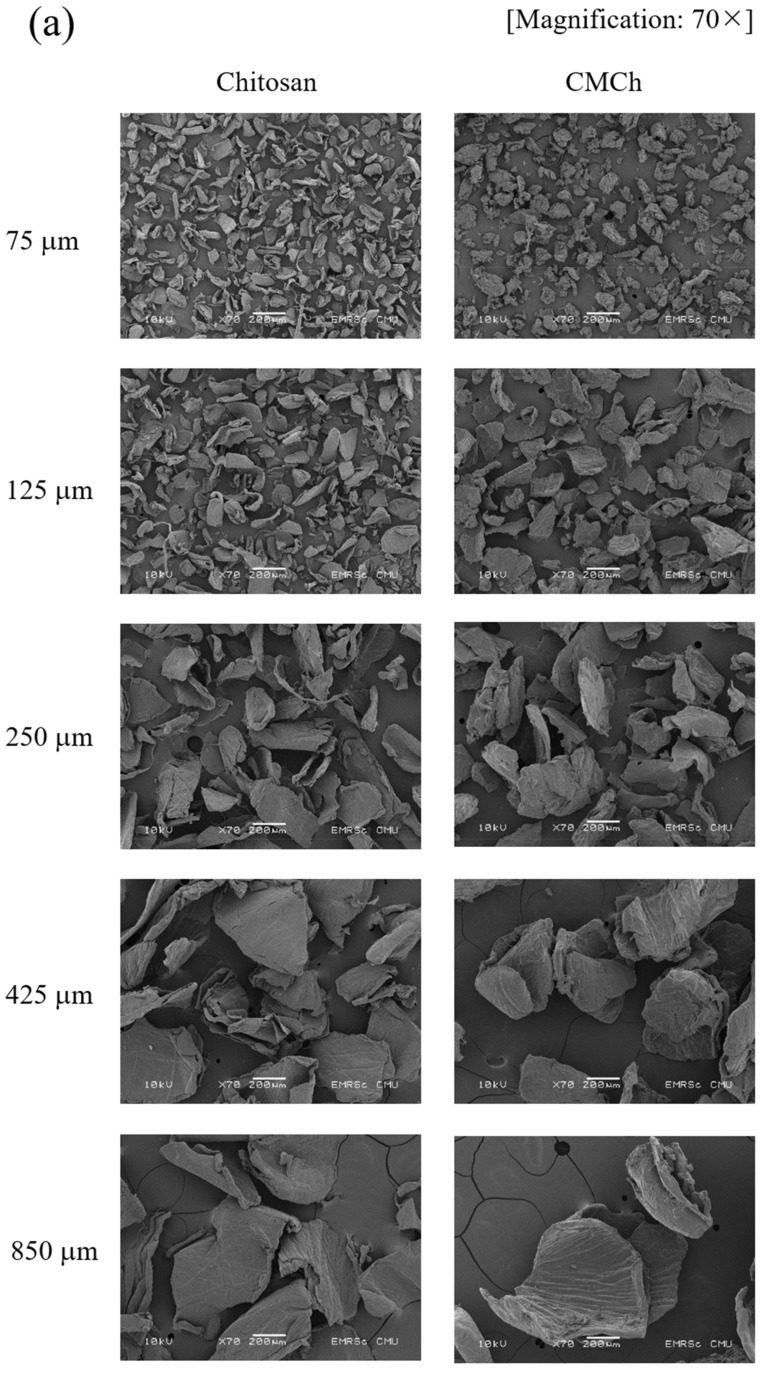
SEM images of chitosan and CMCh powders prepared from chitosan of different particle sizes: (**a**) 70× and (**b**) 1000× magnifications.

**Figure 4 molecules-26-06013-f004:**
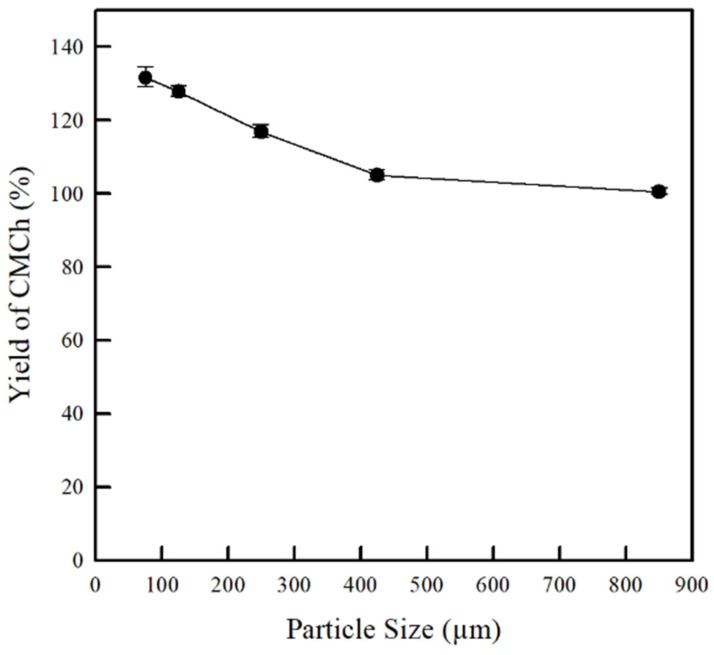
Effect of chitosan particle sizes on yield of CMCh powders.

**Figure 5 molecules-26-06013-f005:**
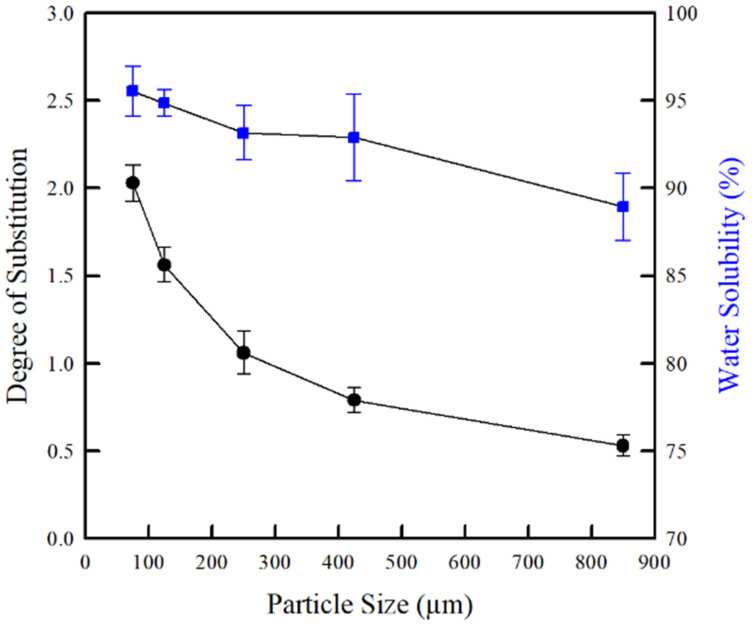
Effect of chitosan particle sizes on degree of substitution and water solubility of CMCh powders.

**Figure 6 molecules-26-06013-f006:**
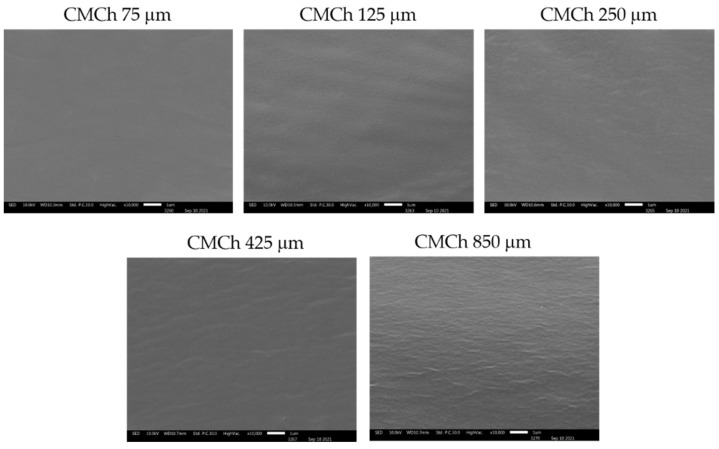
Morphology of CMCh powders.

**Figure 7 molecules-26-06013-f007:**
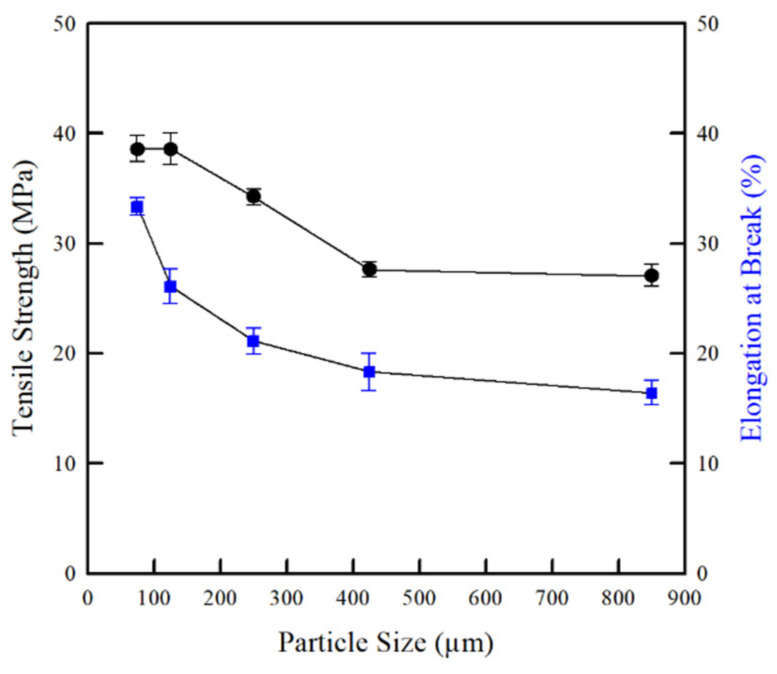
Effect of chitosan particle sizes on tensile strength and elongation of CMCh films.

**Figure 8 molecules-26-06013-f008:**
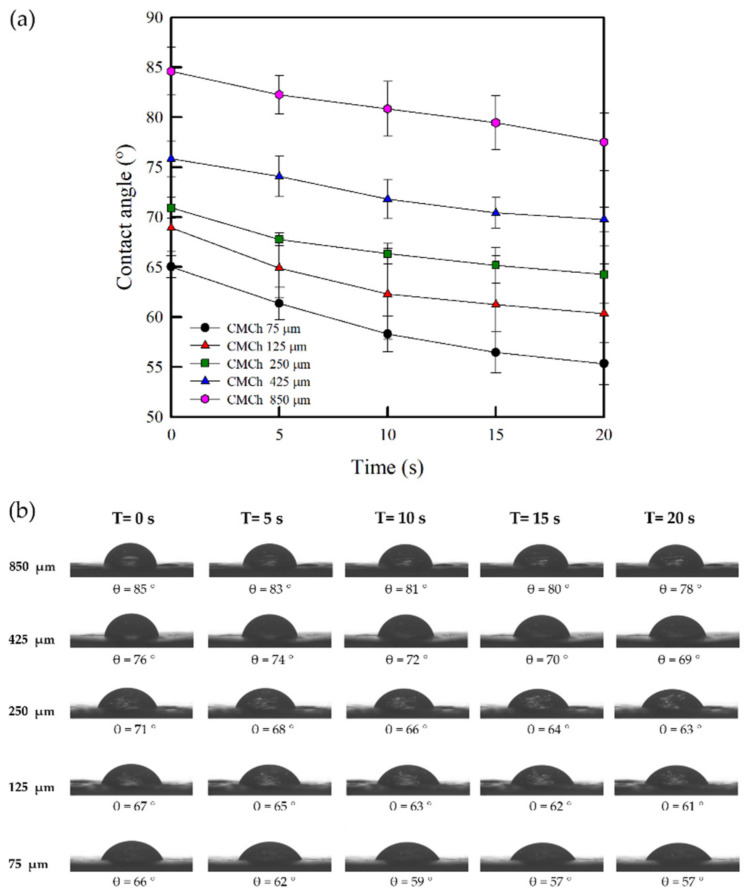
Dynamic contact angle measurement of the CMCh films (**a**) and contact angle images of a 10 µL water droplet on the surface of the CMCh films with times (**b**).

**Figure 9 molecules-26-06013-f009:**
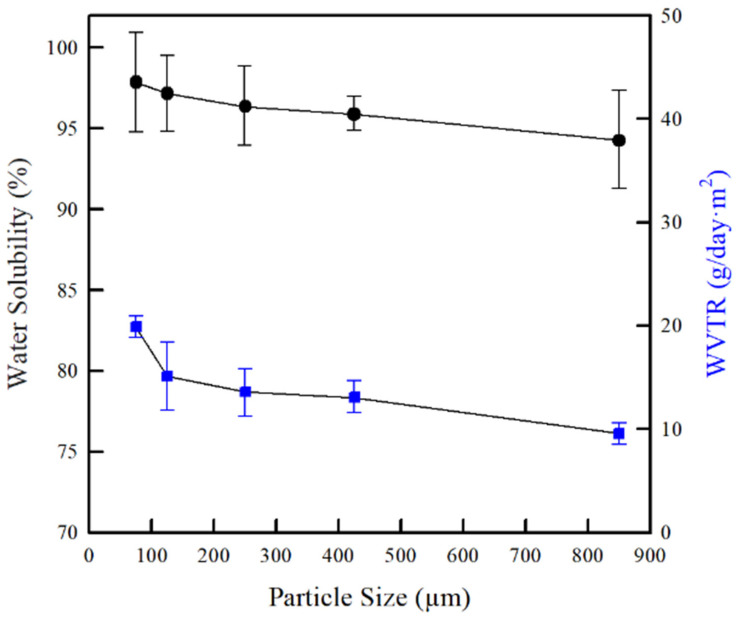
Effect of chitosan particle sizes on water solubility and water vapor transmission rate of CMCh films.

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
