# Peer review of "High Substitution Synthesis of Carboxymethyl Chitosan for Properties Improvement of Carboxymethyl Chitosan Films Depending on Particle Sizes"

_molecules, 2021, doi:10.3390/molecules26196013_

Round 1

Reviewer 1 Report

General comments/questions:

- the discussion should be improved (especially when Ch surface area is indicated) through the SEM analysis of Ch samples. What is the shape of chitosan particles? There should be seen differences not only in size but also in shape.

- the Authors stated that performed modification resulted in higher samples polarity. To prove that, contact angle measurements should be made.

Particular comments:

- line 40 – “The CMCh films with different particle sizes” – this suggests that films can be characterized by different particle sizes – probably should be “The CMCh films prepared with CMCh based on chitosan with different particle sizes were….”

Line 49 – “using chitosan with a smaller size” – chitosan powder can have a size, not chitosan itself.

- lines 44-46 “These phenomena occurred owing to an increase in polarity of the CMCh powder from a higher conversion of chitosan into CMCh. This was also related to a higher surface area in the substitution reaction provided by chitosan with a smaller particle size.”: indicate clearly which properties are related to which one?

- Equation 1 – I have some doubts regarding this equation. As the molar mass of repeating unit od CMCh is higher than Ch, then 25g of Ch (when Yield = 100%) should result in CMCh mass higher than 25g. In my opinion, 25 g of substrate will produce with Yield = 100% more than 25 g of a product. In this context, the equation for Yield calculation is not proper, as, in the situation of 100% conversion, the calculated value will be higher than 100%.
The Law of Conservation of Mass states that matter can neither be created nor destroyed; all that happens is that it changes forms. Therefore, a reaction can never have >100% actual yield (see figure 2). Moreover, using Equation 1 in its present form will give Yield values corresponding strongly to the degree of substitution; thus, data presented in Fig. 2 (decrease in Yield) can be simply a result of decreasing DS value.

- based on the above suggestion, experimental data should be recalculated and used to change Figure 2 and the discussion given in 2.2. Cut also part of the Y-axis (empty area below the curve) to highlight the differences between the particular points.

- Figure 1 – should be higher. Please provide an enlarged figure of the region corresponding to C=O vibration in COOH.

- Provide the chemical structures of both Ch and CMCh (can be in the form of a reaction scheme)

- Line 139: “The chitosan with a smaller particle size has a higher DS” – DS is a property of CMCh, should be “resulted in CmCh of higher DS.”

- Line 164-165 “higher DS. This resulted in an increase in intermolecular force between molecules of CMCh [43], leading to an enhancement of mechanical properties.”: if the number and strength of interchain interaction increases, then the elongation at break and TS should decrease, not increase. Thus higher DS values should cause a higher number of interchain interactions and, finally, lower elongation at break values. How can you explain the results presented in Fig. 4? [consider the changes in the internal structure, density, crystallinity]

- Mechanical properties of CMCh films: The plastification effect of glycerin should also be discussed. TS and Eb values should be compared with those presented in the literature for other CMCh-based films.

- line 190: “In addition, the molecular weight of chitosan affects the WVTR of the CMCh film” – overestimated, a molecular weight of CMCh (connected with the molecular weight of Ch used in CMCh synthesis)

- please compare (add any statement) the water solubility of CMCh powders end films

- the discussion of WVTR should be improved – the differences in the internal film structure should also be considered (can be expressed by density evaluated through weight method). The transport of water and final WVTR values depends on the internal network. Provide if the CMCh films are porous or dense?

Author Response

Reviewer 1

English language and style

( ) Extensive editing of English language and style required
( ) Moderate English changes required
(x) English language and style are fine/minor spell check required
( ) I don't feel qualified to judge about the English language and style

Yes

Can be improved

Must be improved

Not applicable

Does the introduction provide sufficient background and include all relevant references?

(x)

( )

( )

( )

Is the research design appropriate?

( )

( )

(x)

( )

Are the methods adequately described?

( )

(x)

( )

( )

Are the results clearly presented?

( )

(x)

( )

( )

Are the conclusions supported by the results?

( )

( )

(x)

( )

Comments and Suggestions for Authors

General comments/questions:

Question: The discussion should be improved (especially when Ch surface area is indicated) through the SEM analysis of Ch samples. What is the shape of chitosan particles? There should be seen differences not only in size but also in shape.

Answer: As you suggested, we have performed the SEM analysis. In addition, we also performed the stereo microscope to study the morphology and physical appearance of chitosan and CMCh powders. Both chitosan and CMCh powders showed an irregular shape, which can be seen in the stereo microscope (Figure 2) and SEM images (Figure 3 (a,b))  However, the CMCh powders have more transparency than that of chitosan due to the reduction of crystallinity, which has been investigated in our previous study by XRD. The reduction of crystallinity is attributed to the formation of the bulky group on the surface of chitosan after chemical modification. In addition, surface of CMCh powders was rougher than that of chitosan due to the formation of bulky groups (-CH2COOH) on its surface. We also measured the average particle size of chitosan and CMCh powders using SEM analysis. As expected, the width of CMCh particles was wider than that of chitosan particles due to effect of the formation of the bulky groups (-CH2COOH) on the surface of CMCh particles. On the contrary, the length of CMCh particles was shorter than that of chitosan particles due to effect of the chemical reaction during surface modification, leading to chain scission.

Question: The Authors stated that performed modification resulted in higher samples polarity. To prove that, contact angle measurements should be made.

Answer: The CMCh powder with a smaller particle size has a higher DS due to a higher surface area generated by a smaller particle, which led to a greater conversion of chitosan into CMCh. Similar influence of particle size on the DS of CMC has been reported by Rahman et al. [28] and Yeasmin and Mondal [23]. The higher DS implied an increase in polarity of CMCh powders as reflected by a larger number of polar groups (-OH, -NH2, and –COOH) and explained by the FT-IR results

The contact angle results show the same trend. The CMCh with smaller the chitosan particle size has the lower water contact angle (Figure 8). It means that the CMCh with smaller the chitosan particle size has more polarity, which agree with DS and water solubility results (Figure 5).

 Particular comments:

Question:  line 40 – “The CMCh films with different particle sizes” – this suggests that films can be characterized by different particle sizes – probably should be “The CMCh films prepared with CMCh based on chitosan with different particle sizes were….”

Answer: As you commented, we have changed the sentence. It reads now on page 1, “The CMCh films prepared with CMCh based on chitosan different particle sizes were fabricated by a solution casting technique.”

Question: Line 49 – “using chitosan with a smaller size” – chitosan powder can have a size, not chitosan itself.

Answer:  The sentence was changed. It reads now on page 1, “This study demonstrated that a greater improvement in water solubility of the CMCh powders and films can be achieved by using chitosan powder with a smaller size”.

Question: lines 44-46 “These phenomena occurred owing to an increase in polarity of the CMCh powder from a higher conversion of chitosan into CMCh. This was also related to a higher surface area in the substitution reaction provided by chitosan with a smaller particle size.”: indicate clearly which properties are related to which one?

Answer:  The sentence was changed. It reads now on page 1, “The increase in water solubility was owing to an increase in polarity of the CMCh powder from a higher conversion of chitosan into CMCh. In addition, the higher conversion of chitosan was also related to a higher surface area in the substitution reaction provided by chitosan powder with a smaller particle size.”

Question: Equation 1 – I have some doubts regarding this equation. As the molar mass of repeating unit of CMCh is higher than Ch, then 25g of Ch (when Yield = 100%) should result in CMCh mass higher than 25g. In my opinion, 25 g of substrate will produce with Yield = 100% more than 25 g of a product. In this context, the equation for Yield calculation is not proper, as, in the situation of 100% conversion, the calculated value will be higher than 100%.
The Law of Conservation of Mass states that matter can neither be created nor destroyed; all that happens is that it changes forms. Therefore, a reaction can never have >100% actual yield (see figure 2). Moreover, using Equation 1 in its present form will give Yield values corresponding strongly to the degree of substitution; thus, data presented in Fig. 2 (decrease in Yield) can be simply a result of decreasing DS value.

Answer:  As shown in the below equation and schematic reaction, the –CH2COOH or –CH2COONa group in CMCh molecule has a higher molecular weight than that of H in chitosan molecule. Therefore, when the –H was replaced by –CH2COOH or –CH2COONa, the weight of the product automatically increase by itself. As a conversely, the yield of the product also automatically increased, which over 100% calculating by the below equation.      

Yield (%)=W1/W0 ×100

Where, W0 is weight of chitosan (g) and W1 is weight of the obtained CMCh (g).

Question: based on the above suggestion, experimental data should be recalculated and used to change Figure 2 and the discussion given in 2.2. Cut also part of the Y-axis (empty area below the curve) to highlight the differences between the particular points.

Answer:  As explained in the previous question, there is no any mistake in the calculation of the yield and DS. Therefore, we would like to maintain the original Figure 4 and 5 in the revise version of manuscript.

Question: Figure 1 – should be higher. Please provide an enlarged figure of the region corresponding to C=O vibration in COOH.

Answer:  An enlarged figure of the region corresponding to the vibration of COO- group overlapped with the original N–H bond of chitosan at 1599 cm−1 and C=O stretch of carboxylic group at 1741 cm−1 were shown in Figure 1 of the revised version of manuscript.

Question: Provide the chemical structures of both Ch and CMCh (can be in the form of a reaction scheme)

Answer: As you commented, we have included the below reaction in the supplementary materials.

Question: Line 139: “The chitosan with a smaller particle size has a higher DS” – DS is a property of CMCh, should be “resulted in CmCh of higher DS.”

Answer:  The sentence was changed. It reads now on page 1, “The CMCh powder with a smaller particle size has a higher DS due to a higher surface area generated by a smaller particle, which led to a greater conversion of chitosan into CMCh.”

Question: “higher DS. This resulted in an increase in intermolecular force between molecules of CMCh [43], leading to an enhancement of mechanical properties.”: if the number and strength of interchain interaction increases, then the elongation at break and TS should decrease, not increase. Thus higher DS values should cause a higher number of interchain interactions and, finally, lower elongation at break values. How can you explain the results presented in Fig. 4? [consider the changes in the internal structure, density, crystallinity]

Answer: Generally, polymer properties are dependent on regularity of molecular structure, conformational flexibility, intermolecular force, and crystallinity. In this study, as explained, the intermolecular force (i.e., electrostatic) increased after chemical modification, which result in an increase in tensile strength. However, elongation of the CMCh film did not decrease due to the reduction of crystallinity, which results in an enhancement of conformation flexibility, leading to an increase in elongation at break.

Question: Mechanical properties of CMCh films: The plastification effect of glycerin should also be discussed. TS and Eb values should be compared with those presented in the literature for other CMCh-based films.

Answer:  In this study, we did not study the effect of plasticizer. We have fixed the concentration of plasticizer at 30% (w/v). Therefore, the plasticization effect of glycerol did not discuss in the manuscript. We have discussed our TS and EB with other work in the revised version of manuscript.

Question: “In addition, the molecular weight of chitosan affects the WVTR of the CMCh film” – overestimated, a molecular weight of CMCh (connected with the molecular weight of Ch used in CMCh synthesis)

Answer: “In addition, the molecular weight of chitosan affects the WVTR of the CMCh film, in which the CMCh film prepared from the chitosan with higher molecular weight exhibits a lower WVTR compared to those with chitosan with a lower molecular weight [44].” This sentence is the results of other research group. We would to provide the principle of the relationship between molecular weight of polymer and WVTR of CMCh films to the reader.

Question: please compare (add any statement) the water solubility of CMCh powders dnd films

Answer:  As you suggested, we have compared the water solubility of CMCh powders and CMCh films in the revised version of manuscript. This sentence “Notably, the water solubility of CMCh films was relatively higher than that of CMCh powders due to the influence of hydrophilic character of plasticizer (glycerol) containing the CMCh film.” was added.

Question: the discussion of WVTR should be improved – the differences in the internal film structure should also be considered (can be expressed by density evaluated through weight method). The transport of water and final WVTR values depends on the internal network. Provide if the CMCh films are porous or dense?

Answer:  As shown in the SEM images (Figure 6), there was no internal pore in the fractured surface of all CMCh films. This indicated that the WVTR of the CMCh films mainly relies on their polarity or hydrophilic character, and crystallinity as explained in the manuscript.  

Reviewer 2 Report

The authors present and discuss the synthesis of carboxymethyl chitosan and the preparation of the film depending on the particle size of pristine chitosan.
The authors demonstrate that the degree of substitution and film properties depend on the size of the chitosan particles separated by mechanical procedures.
I find interesting the solubility increase related to the particle size as well as to the degree of substitution which could be of relevant interest for application purposes by extending the possibility of using the polysaccharide in milder pH conditions.
I believe that the conclusions are well supported by the experiments conducted even if the correlation reactivity - particle size is well known for a pletora of nanosystems as based on fundamental concepts of thermodynamics.
Overall, the manuscript is clearly written, the conclusions are supported by the experiment but, in my opinion, the manuscript could be improved if the authors can support the evidence with particle size analysis. Nominally, some SEM experiments on the various separate fractions and some dynamic light scattering experiments could be of considerable support to validate the nominal size of the particles and their polydispersion.
I suggest also indicating the brand of monochloroacetic acid.

Author Response

Answer to Comment Reviewer 2

English language and style

( ) Extensive editing of English language and style required
(x) Moderate English changes required
( ) English language and style are fine/minor spell check required
( ) I don't feel qualified to judge about the English language and style

Yes

Can be improved

Must be improved

Not applicable

Does the introduction provide sufficient background and include all relevant references?

( )

(x)

( )

( )

Is the research design appropriate?

(x)

( )

( )

( )

Are the methods adequately described?

(x)

( )

( )

( )

Are the results clearly presented?

(x)

( )

( )

( )

Are the conclusions supported by the results?

( )

(x)

( )

( )

Comments and Suggestions for Authors

Question: The authors present and discuss the synthesis of carboxymethyl chitosan and the preparation of the film depending on the particle size of pristine chitosan.
The authors demonstrate that the degree of substitution and film properties depend on the size of the chitosan particles separated by mechanical procedures.
I find interesting the solubility increase related to the particle size as well as to the degree of substitution which could be of relevant interest for application purposes by extending the possibility of using the polysaccharide in milder pH conditions.
I believe that the conclusions are well supported by the experiments conducted even if the correlation reactivity - particle size is well known for a pletora of nanosystems as based on fundamental concepts of thermodynamics.
Overall, the manuscript is clearly written, the conclusions are supported by the experiment but, in my opinion, the manuscript could be improved if the authors can support the evidence with particle size analysis. Nominally, some SEM experiments on the various separate fractions and some dynamic light scattering experiments could be of considerable support to validate the nominal size of the particles and their polydispersion.

Answer: The authors thank the reviewer for reviewing this manuscript and for your comments and suggestions, which greatly improved the quality of our paper. It took us more than one month to run more experiment. However, we believe the quality of our manuscript significantly improved.

As you commented, we have performed the SEM, stereo microscope, and contact angle. Unfortunately, we do not have dynamic light scattering instrument to validate the particle size. However, the particle size of the chitosan and CMCh samples were directly measured from SEM micrograph. The average particle size of the chitosan and CMCh powders can be obviously seen through the SEM micrograph and picture from stereo microscope. We believe that reviewer and reader can clearly see the different of size. The range and average of chitosan and CMCh particle size were shown in Table 1S.

Questions: I suggest also indicating the brand of monochloroacetic acid.

Answer: This sentence was added “Monochloroacetic acid was purchased from Sigma-Aldrich, Burlington, MA, USA” in 3.1 Materials with yellow highlight.

Submission Date

11 July 2021

Date of this review

01 Aug 2021 19:57:39

Round 2

Reviewer 1 Report

I have read carefully the explanations given by the Authors. The manuscript has been improved from the previous version due to the revisions made by the Authors. They have attended to all my comments by modifying the text accordingly or arguing my statement and making their point clear.